# Development of a Time-Resolved Fluorescent Microsphere Test Strip for Rapid, On-Site, and Sensitive Detection of Picoxystrobin in Vegetables

**DOI:** 10.3390/foods13030423

**Published:** 2024-01-28

**Authors:** Junjie Chen, Lidan Chen, Yongyi Zhang, Siyi Xiang, Ruizhou Zhang, Yudong Shen, Jiaming Liao, Huahui Xie, Jinyi Yang

**Affiliations:** 1Guangdong Provincial Key Laboratory of Food Quality and Safety, College of Food Science, South China Agricultural University, Guangzhou 510642, China; junjiechen361@163.com (J.C.); 13059154257@163.com (L.C.); zhangyongyi@stu.scau.edu.cn (Y.Z.); xiangsiyi0926@163.com (S.X.); zhangruiz1996@163.com (R.Z.); shenyudong@scau.edu.cn (Y.S.); q719438379@126.com (J.L.); 19856935169@163.com (H.X.); 2Wens Institute, Wens Foodstuff Groups Co., Ltd., Yunfu 527400, China

**Keywords:** picoxystrobin, monoclonal antibody, time-resolved fluorescence immunochromatographic assay, vegetables

## Abstract

Picoxystrobin (PIC) is a fungicide extensively used for disease control in both crops and vegetables. Residues of PIC in vegetables pose a potential threat to human health due to their accumulation in the food chain. In this study, a specific PIC monoclonal antibody (mAb) was developed by introducing a carboxylic acid arm into PIC and subsequently preparing a hapten and an artificial antigen. A sensitive and rapid time-resolved fluorescence immunochromatographic assay (TRFICA) was established based on the mAb. Subsequently, using a time-resolved fluorescent microsphere (TRFM) as signal probe, mAbs and microspheres were covalently coupled. The activated pH, the mAb diluents, the mAb amount, and the probe amount were optimized. Under optimized conditions, the quantitative limits of detection (qLOD) of PIC in cucumber, green pepper, and tomato using TRFICA were established at 0.61, 0.26, and 3.44 ng/mL, respectively; the 50% inhibiting concentrations (IC_50_) were 11.76, 5.29, and 37.68 ng/mL, respectively. The linear ranges were 1.81–76.71, 0.80–35.04, and 8.32–170.55 ng/mL, respectively. The average recovery in cucumber, green pepper, and tomato samples ranged from 79.8% to 105.0%, and the corresponding coefficients of variation (CV) were below 14.2%. In addition, 15 vegetable samples were selected and compared with the results obtained using ultra-performance liquid chromatography/tandem mass spectrometry (UPLC-MS/MS). The results revealed a high degree of concordance between the proposed method and UPLC-MS/MS. In conclusion, the devised TRFICA method is a valuable tool for rapid, on-site, and highly sensitive detection of PIC residues in vegetables.

## 1. Introduction

Picoxystrobin (PIC) is a methoxyacrylate fungicide with high persistence and potent bactericidal activity that is extensively applied to rice, vegetables, and fruits [1]. With the increasing use of PIC in agriculture and the food industry, there is a growing concern about the possibility of exceeding permissible residue levels (the maximum residue limit (MRL) of picoxystrobin in GB2763-2021 [2]) in food [3]. Although this pesticide does not pose a high risk of direct toxicity, prolonged consumption of food with excessive residue levels by humans and animals may entail certain potential risks [4,5]. Previous studies have indicated that when humans and other animals (such as cockroaches and zebrafish) are exposed to trace amounts of this fungicide, adverse health effects may occur, including eye and respiratory tract irritation, weakness, dizziness, skin redness, and chest pain [6,7,8]. Prolonged exposure can lead to malformations, tumors, and mutations [9]. Considering the potential hazards of picoxystrobin, China has established clear guidelines on the maximum residue limit (MRL) of picoxystrobin in some vegetables, such as 0.5 mg/kg for cucumber and pepper and 1 mg/kg for tomato (GB2763-2021). Therefore, it is necessary to establish an effective method for the detection of picoxystrobin residues in vegetables.

Currently, instrumental methods, including gas chromatography-tandem mass spectrometry (GC-MS/MS) [10,11] and liquid chromatography-tandem mass spectrometry (LC-MS/MS) [12,13], are the primary approaches used for detecting picoxystrobin. These methods are widely used because of their high sensitivity, specificity, and accuracy [14]. However, complex pretreatment procedures and instrument operations make it challenging to meet the requirements for rapid on-site sample detection. In recent years, immunoassay-based detection methods have been extensively studied because of their high specificity, sensitivity, and simplicity [15,16]. The detection limit of the enzyme-linked-immunosorbent serologic assay (ELISA) method for picoxystrobin developed by Mercader and Esteve reached 0.1 ng/mL and 0.2 ng/mL, respectively [17,18]. Although their ELISA method achieved a low detection limit, it still did not resolve the problems of prolonged reaction time and tedious washing steps inherent to ELISA, rendering it unsuitable for on-site rapid detection. With the development of immunodetection technology, immunochromatographic assays (ICAs) combine the separation ability of chromatography with the high specificity of traditional immunoassay methods, making them particularly suitable for real-time on-site detection [19,20]. Therefore, this is an inexpensive and portable monitoring tool that can be used for various diagnostic purposes. In the construction of ICA, gold nanoparticles are commonly used as labeled materials; however, their labeled products are not sufficiently stable and have low sensitivity, allowing for only qualitative and semi-quantitative detection [21]. For example, the PIC immunocolloidal strip developed by Song et al. has a detection limit of 100 ng/mL, low sensitivity, and is unable to meet the market demand for trace PIC detection [22]. Time-resolved fluorescent microsphere (TRFM), a novel type of labeling material, possesses unique fluorescence characteristics that amplify signals and enhance sensitivity [23]. This topic has become a research hotspot in recent years [24,25]. Compared to other marker materials, TRFMs exhibit higher fluorescence intensity, longer fluorescence lifetime, larger Stokes shift, and smaller marker volume. The fluorescence decay time of TRFMs is 103–106-fold that of other fluorescent materials, effectively reducing nonspecific signals and achieving high sensitivity and stability [26,27]. However, time-resolved fluorescence immunochromatographic assay (TRFICA) has not been reported in the detection of PIC.

In this study, utilizing the acryloyl methyl ether on the PIC molecule as the starting point, a carboxylic acid arm was introduced through a reaction with 6-bromohexanoic acid, preparing the PIC hapten. Subsequently, the synthesized hapten was coupled with a carrier protein to serve as an immunogen, and a monoclonal antibody that specifically recognizes picoxystrobin was obtained from Balb/c mice. Based on this antibody, a time-resolved fluorescence immunochromatographic assay method with a competitive reaction mode was established using time-resolved fluorescent microspheres as a tracer. The method was evaluated from the aspects of quantitative standard curve, detection limit, linear range, sensitivity, specificity, accuracy, and recovery rate, and then applied for the analysis of blind vegetable samples.

## 2. Materials and Methods

### 2.1. Chemicals and Reagents

Picoxystrobin standards and 6-bromohexanoic acid (≥99%) were purchased from McLean Biochemical Co., Ltd. (Shanghai, China). Fluoxastrobin (≥98%), imazalil (≥98%), azoxystrobin (≥98%), cyproconazole (≥98%), dimoxystrobin (≥98%), pyraclostrobin (≥98%), and kresoxim-methyl (≥98%) were acquired from Dr. Ehrenstorfer (Augsburg, Germany). Bovine serum albumin (BSA), ovalbumin (OVA), keyhole limpet hemocyanin (KLH), lactoferrin (LF), goat anti-mouse IgG, horseradish peroxidase (HRP)-labeled goat anti-mouse IgG, 2-(N-Morpholino) ethanesulfonic acid (MES), tris (hydroxymethyl)aminomethane, and complete and incomplete Freund’s adjuvants were obtained from Sigma-Aldrich (St. Louis, MO, USA). N-hydroxysuccinimide (NHS, 99%), 1-(3-dimethylaminopropyl)-3-ethylcarbodiimide (EDC, 99%), and N,N-dimethylformamide (DMF) were purchased from Aladdin Chemistry Co., Ltd. (Shanghai, China). Time-resolved fluorescence microspheres (1% (*w*/*v*) solid suspension) were sourced from Bangs Laboratories, Inc. (Fishers, IN, USA). PSA and C_18_ purchased from Yuexu Technology Co., Ltd. (Shanghai, China). All the other conventional reagents and chemicals were of analytical grade or higher.

### 2.2. Instruments and Materials

The absorbent pad (CH37K), PVC backing plate (SMA31-40), and sample pads (HG-2) were purchased from Shanghai Liangxin Co., Ltd. (Shanghai, China). The nitrocellulose (NC) membranes (UniSartCN140) were purchased from Sartorius Stedim Biotech GmbH (Goettingen, Germany). A Zetasizer Nano ZS90 (Malvern Panalytical, Malvern, UK) was used to measure the size and charge of the nanoparticles. A UniqueR-10 pure water instrument was purchased from Ewell Bio-Technology Co., Ltd. (Guangzhou, China). A microplate reader (MultiskanMK3) was purchased from Thermo Fisher Scientific (Waltham, MA, USA). The UV analyzer was obtained from Zhicheng Technology Co., Ltd. (Zhengzhou, China). A DEM-3 automatic washing machine was purchased from Beijing Tuopu Analytical Instrument Co., Ltd. (Beijing, China). An ultralow-temperature high-speed centrifuge was purchased from Eppendorf Inc. (Hamburg, Germany). A BioDot-XYZ 3060 Dispensing Platform was purchased from BioDot Inc. (Irvine, CA, USA). A programmable strip cutter ZQ-2000 was supplied by Shanghai Kinbio Tech Co., Ltd. (Shanghai, China). A fluorescence immunity analyzer was supplied by Nanjing Microdetection Technology Co., Ltd. (Nanjing, China).

### 2.3. Preparation of Hapten and Artificial Antigen

Hapten design is a pivotal step towards achieving highly sensitive monoclonal antibodies. Ensuring adequate exposure of the small molecule structure during the design process is essential for immune effects [28]. The hapten was synthesized using a slight modification of a previous report [22]. The route of PIC hapten synthesis is shown in Figure 1A. Briefly, LiOH (593 mg, 14.1 mM) and Bu_4_NHSO_4_ (479 mg, 1.41 mM) were dissolved in a 14 mL THF/H_2_O (3:1) solution. PIC (518 mg, 1.41 mM) was added to the solution at 25 °C. The solvent was removed via rotary evaporation to obtain an intermediate product. Next, 414 mg (1.5 mM) of 6-bromohexanoic acid was dissolved in 2 mL of DMF. The intermediate product (500 mg, 1.0 mM), anhydrous K_2_CO_3_ (391 mg, 2 mM), and triethylamine (295 μL, 1.5 mM) were dissolved in DMF. After removing water with anhydrous Na_2_SO_4_, rotary evaporation of DMF yielded a yellowish oily substance. The oily substance was purified using silica column elution (EA:PE = 1:1), followed by rotary evaporation of the organic solvent, resulting in the final product, the PIC hapten (PIC-H). PIC-H was confirmed by electrospray ionization mass spectrometry (ESI-MS).

The immunogen was obtained by coupling the hapten with carrier proteins using the active ester method (Figure 1B). Specifically, the hapten PIC-H (27 mg, 1.0 mML), 1-(3-dimethylaminopropyl)-3-ethylcarbodiimide (EDC, 16.63 mg, 1.5 mM), and N-hydroxysuccinimide (NHS, 9.97 mg, 1.5 mM) were dissolved in 800 μL DMF and stirred at 4 °C overnight. Ten milligrams of carrier proteins (OVA, BSA, KLH, and LF) were dissolved in 3 mL of 0.1 M PBS buffer (pH = 7.9). Subsequently, the hapten-activated solution was added dropwise to the carrier protein solution and stirred at 4 °C for 12 h. The reactive mixture was also dialyzed against PBS (0.01 M, pH 7.4) at 4 °C for 72 h, and stored at −20 °C. The obtained hapten–protein conjugates were confirmed by UV–vis spectrometry.

### 2.4. Preparation and Characterization of PIC Monoclonal Antibody

Healthy 5-week-old female Balb/c mice were subcutaneously immunized every two weeks with a mixture of 200 μL immunogen and Freund’s adjuvant (Figure 2A). One week after the third immunization, antisera were collected from the mouse tail for sensitivity testing by icELISA. Four consecutive immunizations were conducted, and mice with the highest potency and optimal inhibition rate were selected for cell fusion [29]. A boost immunization was performed three days before cell fusion by injecting 200 μL of the immunogen directly into the mouse abdominal cavity. The spleen cells of Balb/c mice with the highest efficacy and inhibition were observed in antiserum determination and fused with Sp2/0 myeloma cells under the action of PEG2000 [30]. PIC hybridoma cell lines capable of stably and uniformly secreting antibodies were screened using the limiting dilution method. The chosen hybridoma cell line was injected into the abdominal cavity of Balb/c mice to induce ascites, and the generated ascites were purified by G protein affinity chromatography to obtain PIC mAbs. A mouse monoclonal antibody subtype identification kit was used to subtype the mAb in ascites, and the purity of the mAb was assessed by SDS-PAGE.

### 2.5. TRFM-Labeled PIC Antibody and Its Characterization

Activated TRFMs were covalently coupled to a PIC mAb to prepare TRFM immunoprobes. In brief, 200 μL of MES (0.05 M) and 10 μL of TRFMs (1% *w*/*v*) were mixed and dispersed using ultrasound. Following centrifugation (15,000 rpm, 15 min), 200 µL of MES was added for resuspension. Subsequently, 30 µL of NHS (0.1 mg/mL) and 25 µL of EDC (0.1 mg/mL) were added to activate the carboxyl groups on the TRFMs. The mixture was activated in an oscillator at 600 rpm for 15 min and centrifuged at 15,000 rpm for 15 min, and the precipitate was collected. The precipitate was combined with 200 µL of boric acid buffer (BB, 0.05 M, pH = 8.0) and an optimized volume of PIC mAb for coupling with the activated TRFMs. The mixture was shaken at 25 °C for 20 min. Finally, 1 mL of the blocking solution was added to close the active sites on a shaker at 25 °C, 600 rpm for 20 min. The TRFM-mAb complex was centrifuged at 15,000 rpm for 15 min to remove the unreacted antibodies and BSA. After ultrasound resuspension, the sediment was dissolved in 200 μL of phosphate buffer (PB, 0.5 M, pH 7.4) and kept at 4 °C. The fluorescent probes before and after coupling were characterized by transmission electron microscopy and zeta potential measurements.

### 2.6. TRFICA Test Strip Assembly

The lateral flow immunoassay strip consisted of a PVC base, a nitrocellulose (NC) membrane, a sample pad, and an absorbent pad. The coated antigen and goat anti-mouse IgG were diluted to 0.41 mg/mL and 0.08 mg/mL, respectively, with PBS solution (0.01 M, pH 7.4). The coated antigen was sprayed on the detection line (T line), goat anti-mouse IgG was sprayed on the quality control line (C line), and the spray volume was set to 0.9 μL/cm. As depicted in Figure 2B, the NC membrane was affixed to the middle of the PVC base, and the absorbent pad and sample pad were affixed to the upper and lower ends of the PVC base, with a 2 mm overlap with the NC membrane. The assembled test strip was cut into 3.6 mm wide strips and stored in a dry cabinet for subsequent use.

### 2.7. TRFICA Detection Procedure and Principle

The lateral-flow immunoassay strip used a competitive reaction mode (Figure 2B) [31]. The PIC antigen and anti-mouse IgG were affixed to the NC membrane. During the detection process, the analyte in the sample bound to the TRFM probe in the microhole, inhibiting the binding between the TRFM probe and the antigen on the NC membrane and resulting in the coloration of the T line, indicating a positive result. When the TRFM probe bound to the antigen on the NC membrane through lateral flow, it caused simultaneous coloration of the T and C lines, indicating a negative result. When line C remained colorless, the result was considered invalid. The T line fluorescence intensity exhibited a negative correlation with the PIC concentration in the test solution. An optimized volume of fluorescence microsphere-labeled antibodies (TRFM-PIC-mAbs) and 100 μL of standard or test sample were added to the microhole. After a 5 min incubation at room temperature, the test strip was inserted into the microhole. Following an 8 min incubation at room temperature, the test strip was removed, the coloration was assessed under ultraviolet light, and the T- and C-line fluorescence intensities were collected using a fluorescence immunity analyzer to perform qualitative and quantitative analyses of the test samples (Figure 2C).

### 2.8. Performance of TRFICA

To attain the optimal fluorescence intensity and sensitivity, the experimental factors affecting TRFICA performance were optimized. Specifically, adjustments were made to the labeling buffer pH (using 0.05 M MES at pH 5.0, 6.0, 7.0, and 8.0) to enhance the preparation efficiency and the antibody activity of TRAM immunoprobes. Five antibody dilution buffers, including 0.01 M PB (PH 7.5), 0.05 M PB, 0.05 M Tris-HCl (PH 7.5), 0.01 M BB (PH 8.0), and 0.05 M BB were used to dilute the mAbs, maintaining their activity and improving the reaction efficiency between mAbs and fluorescence microspheres. PIC mAbs were diluted to 1 mg/mL using the optimized mAbs dilution solution, and different antibody amounts (4, 6, 8, and 10 μL) were coupled with the fluorescence microspheres. During sample detection, TRFM probes (3, 4, 5, and 6 μL) were added for the test strip experiment.

The specificity, expressed as cross-reactivity (CR), was determined by assessing the recognition of seven functional and structural analogues (e.g., fluoxastrobin, imazalil, and azoxystrobin). CR (%) was expressed as the percentage of the IC50 value of the target analyte compared to that of the analogue. The sensitivity of TRFICA analysis was assessed using the quantitative limit of detection and a standard curve. Under the aforementioned optimal conditions, a standard curve was established using the numerical values of the drug concentrations as the x-axis and B/B0 (where B0 is the T/C value without the addition of the drug and B is the T/C value with the addition of the drug) as the y-axis. The quantitative detection limit (qLOD) of the instrument was defined as B/B0 at 90% [32].

### 2.9. Recovery Test and Method Evaluation

Fifteen samples of cucumber, bell pepper, and tomato were randomly selected from the market and pretreated for PIC detection. Specifically, 5.0 g of the sample, 1.5 g of NaCl, and 2.0 g of MgSO_4_ were added to 8 mL of acetonitrile, vortexed for 3 min, and centrifugated at 4000 rpm for 5 min. The resulting supernatant (6 mL) was combined with 500 mg primary secondary amine (PSA) sorbent and 250 mg octadecylsilane chemically bonded silica (C_18_), vortexed for 3 min, and centrifuged at 4000 rpm for 3 min. The supernatant was collected, and the dilution multiple of the test strip with fluorescence intensity close to the optimal standard dilution solution (0.05 M BB PH8.0) was chosen to eliminate the sample matrix. TRFICA accuracy was determined based on the average recovery rate and the coefficient of variation. To verify the accuracy of TRFICA, three levels of PIC were spiked into negative samples of cucumber, bell pepper, and tomato after preprocessing. Each level was tested in triplicate (*n* = 3) with the TRFICA test strips and the average recovery rate and coefficient of variation were calculated based on the results. The coefficient of variation (CV%) was computed according to the following formulae: recovery (%) = measure value/spiked value) × 100% and CV (%) = (standard deviation/average value × 100%. Following TRFICA detection, UPLC-MS/MS was used for verification. The UPLC-MS/MS was carried out by electrospray ionization (ESI) positive ion source [33]. The extract supernatant samples were analyzed on Thermoelectric Accucore AQ column (150 mm × 2.1 mm, 2.6 μm) and the temperature was 40 °C, with a mobile phase consisting of a methanol solution (A) and a water solution (B). The elution program was as follows: 0–7 min, 88% B–52% B; 7–16 min, 52% B–20% B; 16–17 min, 20% B–0% B; 17–18 min, 0%–0% B; 18–21 min, 0%–88% B. The injection volume and flow rate were 8 μL and 0.3 mL/min, respectively.

## 3. Results and Discussion

### 3.1. Synthesis of Haptens and Antigens

Due to its small molecular weight, PIC lacks immunogenicity and cannot directly stimulate an animal organism to produce a specific antibody. Conjugation with carrier proteins (OVA/BSA/KLH/LF) enables indirect induction of B cell proliferation and differentiation using T cell epitopes, thereby generating a specific antibody [34]. However, PIC lacks functional groups that directly bind to carrier proteins, and active groups must be introduced into the haptens. Hapten derivatives produced by introducing active groups must maintain a structure, electron distribution, and hydrophobicity similar to those of the original haptens to avoid the production of arm antibodies [35,36]. Therefore, finding a spacer arm that could increase molecular weight without inducing the production of arm antibodies became particularly important. With this foundation, we took 3-methoxy-1-propene on the PIC as the starting point, and a carboxylic acid arm was introduced through a reaction with 6-bromohexanoic acid. The modified hapten maintained the characteristic structure of the original molecule, which allowed for the immune-active cells of animals to recognize the hapten conjugates to the maximum extent.

The target molecular weight of the hapten is 467.17 Da, as indicated by the ESI-MS results. In negative ion mode (Appendix A) and positive ion mode (Appendix A), fragments of 466.0 Da and 468.1 Da were detected, confirming the successful synthesis of the hapten. For the immunogens and coating antigens, the structural features of the carrier protein were added based on the hapten. The coupling curve (Appendix A) illustrates changes in the characteristic peaks of the hapten, carrier protein, and conjugated protein between 200 nm and 350 nm. Appendix A demonstrates that the characteristic peak of the conjugated protein broadens between 250 nm and 280 nm, with a noticeable shift in the peak position. These results indicate the successful synthesis of both the hapten and the artificial antigen.

### 3.2. Preparation and Identification of PIC-mAb

The antisera test results show (Figure 3A) that the mice immunized by three kinds of immunogen all have an immunogenic effect. Compared to the other two immunogens, the PIC-H-KLH-immunized mouse antiserum analysis demonstrated optimal sensitivity and antibody affinity, with the highest titer and inhibition rate (1:32,000, 81%), and was subsequently selected for fusion. After fusion and limited dilution, the monoclonal positive cell lines PIC 14 and PIC 16 with high sensitivity to PIC were obtained and their performance was analyzed. As shown in Figure 3B, the monoclonal positive cell line PIC 16 with a lower IC_50_ was used to produce PIC-mAb. The mAb was purified using protein G affinity chromatography, and subtype determination revealed that PIC 16 cell lines primarily produced IgG1 type mAb (Appendix A). SDS-PAGE under reducing conditions can break the disulfide bonds in antibody proteins and yield light chains (22 kDa) and heavy chains (55 kDa) of mAbs. Therefore, the purification effect was confirmed by SDS-PAGE. SDS-PAGE (Figure 3C) confirmed the purification effectiveness of the PIC 16 mAb, distinctly revealing bands corresponding to the heavy and light chains of the PIC mAb, attesting to its high purity.

### 3.3. Characterization of the TRFM-mAb Probe

TRFMs were coupled with an antibody by covalent interactions, which achieved a close combination of the antibody and microspheres. This induced alterations in the particle size and ζ-potential of the probes after conjugation [37]. To obtain uniform and stable TRFM immunoprobes, TRFMs used to modify monoclonal antibodies should have uniform particle size and uniform dispersity. As shown in Figure 4A,C, the TRFMs’ particles exhibit good dispersion, uniform shape, and a mean diameter of about 255 nm. Following the successful coupling of the TRFM-mAb probe, the average particle size increased to 396 nm (Figure 4B,D), with the ζ-potential transitioning from −29.2 mV to −13.1 mV (Figure 4E). These results indicate that the preparation of the TRFM-mAb probe was successful.

### 3.4. Optimization of TRFICA Working Conditions

During surface carboxylation, the fluorescent microspheres covalently couple with amino groups on the antibody surface, in which EDC and NHS could effectively activate hydroxyl groups, promoting the formation of ester bonds [20]. The pH value during the activation of fluorescent microspheres can influence the preparation efficiency of the fluorescent probe and the antibody activity, consequently affecting the reaction efficiency between the fluorescent microspheres and the antibody. In the labeling process, the activation pH was adjusted to 5, 6, 7, and 8 with MES (0.05 M) solution. The result indicates that, with increasing pH, the inhibition rate exhibits an initial rise followed by a subsequent decline. At the higher activation pH, residual probes during the reaction caused severe background interference, and the inhibition rate was low. When the pH was 6.0, the fluorescence intensity of the strip and the inhibiting rate were ideal. Therefore, we chose pH 6.0 as the optimal pH value for the activation buffer (Figure 5A).

The selection of antibody diluents affects the stability and sensitivity of immunoassay methods [38]. To enhance the coupling efficiency and probe stability, five different antibody diluents were used to dilute the antibody. The results (Figure 5B) showed that the resulting fluorescence intensity decreased when BB (pH 8.0) was used as the antibody diluent. This may be because the alkaline environment of the BB buffer reduced the activity and stability of antibody–microsphere conjugates, thus reducing the fluorescence intensity. Although the inhibition rate of 0.01 M PB (pH 8.0) was not the highest, it was still selected as the antibody diluent for its superior fluorescence intensity and low background.

An appropriate amount of antibody can achieve good color development and reduce steric hindrance to improve coupling efficiency [39]. As shown in Figure 5C, as the antibody concentration increased, the fluorescence intensities of the C and T lines gradually increased. The highest inhibition rate was achieved with 6 μL (6 μg) of antibodies; nevertheless, at this moment the strip fluorescence intensity was relatively faint. While the amount of antibody was greater than 6 μL (6 μg), the inhibition effect became worse, which indicated that the binding of antibody and microspheres had approached saturation. When the antibody amount increased to 8 μL (8 μg) and 10 μL (10 μg), the strip fluorescence intensity became more pronounced, and the inhibition rate has slightly decreased. Considering sufficient performance and cost savings, the optimal antibody amount was determined to be 8 μL (8 μg).

Similarly, the amount of TRFM probe used affected the color intensity and sensitivity of the test papers. If the amount of TRFM probe is too low, the fluorescence signal might be too weak, whereas an excessively high amount might lead to excessive background staining on the NC membrane, affecting sensitivity. As shown in Figure 5D, the probe amount was optimized in the range of 3–6 μL. With an increase in TRFM probe amount, the sensitivity of the test paper decreased. When the TRFM probe amount was 3 μL, the strip color was too light, and the signal readings were unstable. When the TRFM probe amount was greater than 4 μL, TRFMs probes began to remain on the NC membrane. However, at 4 μL, the test paper exhibited minimal background interference, good color intensity, and the highest inhibition rate. Therefore, 4 μL of the probe was chosen for addition.

### 3.5. Sensitivity and Specificity

Different detection samples may exhibit varying sensitivities due to differences in their matrix. To address this, under optimal conditions, TRFICA PIC-detection standard curves were established for cucumber, green pepper, and tomato samples, assessing the sensitivity of the TRFICA method for different samples. As shown in Figure 6A–C, the IC_50_ of this method for three samples were 11.76, 5.29, and 37.68 ng/mL, respectively; the linear ranges were 1.81–76.71, 0.80–35.04, and 8.32–170.55 ng/mL, respectively; the qLODs were 0.61, 0.26, and 3.44 ng/mL, respectively, which was lower than the MRL of China (GB2763-2021). As shown in Table 1, the LOD proposed by our method was much lower than LODs using instrumental methods. Compared with the two reported PIC immunoassay methods, our method showed a similar LOD and, more importantly, the detection speed was nearly 10-fold faster than them [17,18]. In the sample detection mentioned above, partial variations in qLOD and IC_50_ were revealed between the tomato sample and the other samples, indicating that specific components in tomatoes, such as lycopene, may have caused interference with the matrix. Nevertheless, despite these differences, TRFICA fulfilled the detection criteria for tomato and other samples. Structural and functional analogs of PIC were used to verify the specificity of TRFICA. All drugs were added at the same concentration, and the results in Figure 6D and Appendix A show that when structurally similar compounds were added as the target analyte, the fluorescence intensity of the T line remained essentially consistent. TRFICA exhibited low cross-reactivity (CR < 1.1%) for seven other structurally and functionally similar compounds. The above indicates that the good specificity of TRFICA was confirmed.

### 3.6. Sample Analysis

The accuracy and reproducibility of the developed TRFICA were evaluated by analyzing cucumber, green pepper, and tomato samples spiked with three concentrations of PIC. The recovery rates for the cucumber, green pepper, and tomato samples ranged from 79.8% to 99.6%, 81.8% to 105.0%, and 83.1% to 91.0%, respectively. The coefficients of variation were less than 14.2, 8.5, and 7.5%, respectively (Table 2). These results indicate the good accuracy of the established TRFICA, which is suitable for the rapid detection of PIC in vegetables.

To further validate the established TRFICA method, five samples of cucumber, green pepper, and tomato were randomly purchased from the market, with 5 copies of each. Following preprocessing, the samples were tested using both TRFICA and UPLC-MS/MS methods. The results in Table 3 reveal a detection rate of 13.3% across the 15 samples and show a robust correlation between the detection method and UPLC-MS/MS.

## 4. Conclusions

Initially, we systematically investigated the hapten design, the antigen synthesis, and screened for a sensitive, specific PIC-mAb. To obtain the best detection performance, the experimental parameters of the test strips were optimized, including the activation pH, antibody diluents, and amounts of antibody and TRFM probe. To address potential interferences from different vegetable matrices, we developed detection standard curves for various matrix samples. The corresponding qLOD for cucumber, green pepper, and tomato were 0.61, 0.26, and 3.44 ng/mL, respectively. Blind samples were analyzed by both the TRFICA and UPLC-MS/MS, and a good correlation between the two methods was obtained. This method could be used for rapid screening and detection of blind vegetable samples in the market, providing a means and tool for on-site, rapid, and sensitive monitoring of PIC residues in blind samples.

## Figures and Tables

**Figure 1 foods-13-00423-f001:**
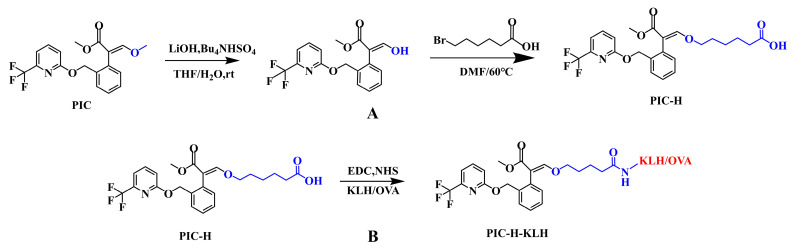
(**A**) Synthetic diagram of PIC-H hapten. (**B**) Hapten–carrier conjugate of PIC-H-KLH/OVA.

**Figure 2 foods-13-00423-f002:**
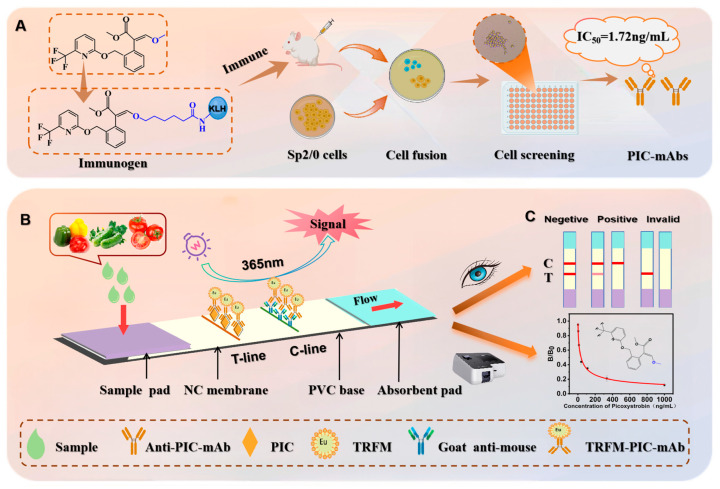
(**A**) Immunity of mice and preparation of TRMS-mAb. (**B**) TRFICA schematic diagram of PIC in cucumber and green pepper. (**C**) TRFICA test results and TRFICA test standard curve.

**Figure 3 foods-13-00423-f003:**
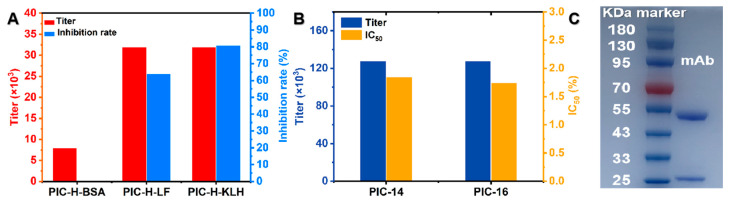
Preparation and characterization of PIC-mAb. (**A**) Immune effects of different antigens. (**B**) Monoclonal positive cell lines performance analysis. (**C**) The SDS-PAGE analysis of PIC-mAb.

**Figure 4 foods-13-00423-f004:**
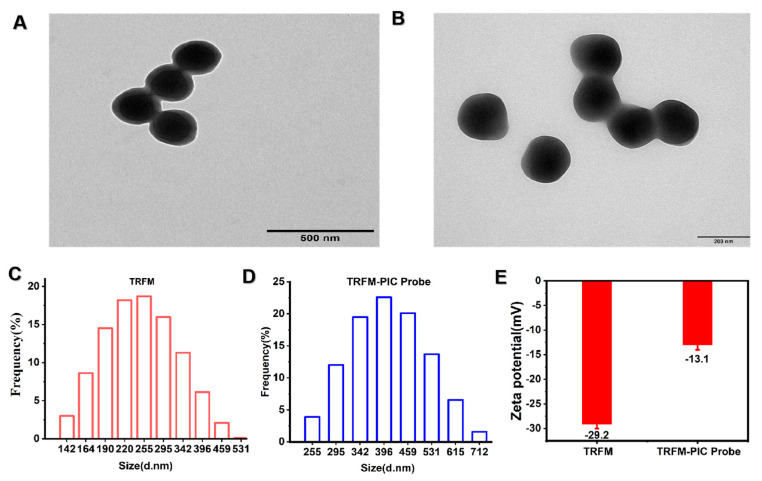
Characterization of fluorescent microsphere antibody labeling. (**A**) TRFM transmission electron microscope image. (**B**) TRFM probe transmission electron microscope image. (**C**) TRFM and TRFM-mAb probe (**D**) particle size, normal distribution. (**E**) Zeta potential of TRFMs and TRFM-mAb probe.

**Figure 5 foods-13-00423-f005:**
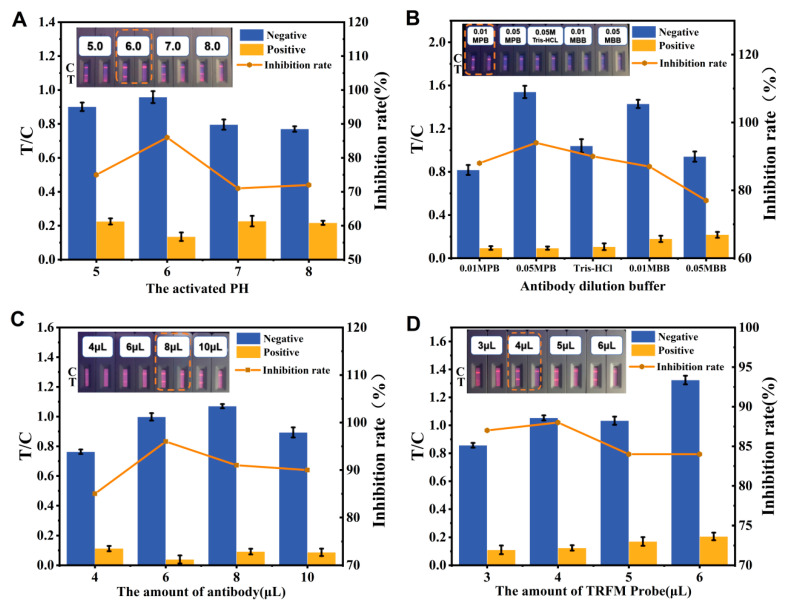
Optimization of TRFICA working conditions. The values of T/C were calculated from the detection results of the above pictures using a fluorescence immunity analyzer and it represents the ratio of T-line and C-line fluorescence intensity. (**A**) Optimization of activation pH. (**B**) Optimization of antibody diluent. (**C**) Optimization of antibody amount. (**D**) Optimization of TRFM probe amount.

**Figure 6 foods-13-00423-f006:**
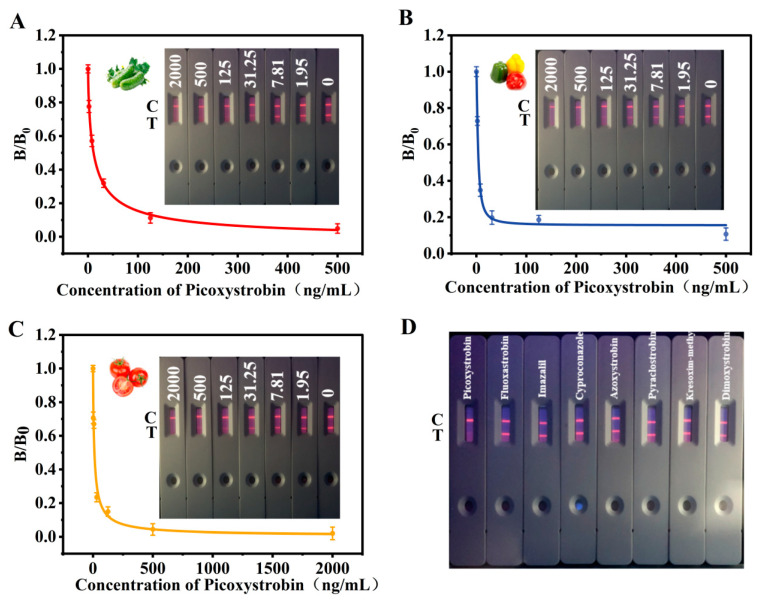
Establishment of naked-eye color diagram and PIC TRFICA standard curve in different samples: (**A**) cucumber, (**B**) green pepper, (**C**) tomato. (**D**) The specificity of TRFICA for picoxystrobin.

**Table 1 foods-13-00423-t001:** Comparison of performance of different methods.

Method	Matrix	The Detection Limit (mg/kg)	References
GC-ECD/ GC-MS	Vegetables, fruits, and crops	0.025	[10]
SPME/GC-MS	Baby food	0.005	[11]
QuEchERS-HPLC	Wheat	0.018	[12]
UPLC-MS/MS	Pepper fruit	0.036	[13]
Ic ELISA	Flour	0.0001	[17]
Ic ELISA	Beer	0.0002	[18]
GICA	Cucumber	0.1	[22]
FPIAs	Red wine	0.005	[16]
TRFICA	Cucumber	0.00026	This Work
	Green pepper	0.00061	
	Tomato	0.0034	

**Table 2 foods-13-00423-t002:** Addition and recovery experiment of TRFICA (*n* = 3).

Sample	Spiked Level (ng/mL)	Detection Level (Mean ± SD) (ng/mL)	Recovery (%)	CV (%)
Cucumber	2	1.60 ± 0.23	79.8	14.2
	12	11.77 ± 1.23	98.1	10.5
	72	71.70 ± 3.05	99.6	4.3
Green pepper	2	1.64 ± 0.11	81.8	6.6
	8	7.34 ± 0.63	91.8	8.5
	32	33.59 ± 1.66	105.0	4.9
Tomato	12	10.52 ± 0.79	87.7	7.5
	42	34.89 ± 2.41	83.1	6.9
	150	136.51 ± 9.13	91.0	6.7

**Table 3 foods-13-00423-t003:** Comparison of picoxystrobin using TRFICA and UPLC-MS/MS in blind samples (*n* = 3).

Assay		TRFICA		UPLC-MS/MS	
Samples	Number	Test Value (Mean ± SD, ng/g)	CV (%)	Test Value (Mean ± SD, ng/g)	CV (%)
Cucumber	Sample 1	ND	-	ND	-
	Sample 2	40.34 ± 5.04	12.4	53.99 ± 6.95	12.8
	Sample 3	ND	-	ND	-
	Sample 4	ND	-	ND	-
	Sample 5	ND	-	ND	-
Green pepper	Sample 1	ND	-	ND	-
	Sample 2	ND	-	ND	-
	Sample 3	ND	-	ND	-
	Sample 4	ND	-	ND	-
	Sample 5	47.90 ± 8.06	16.8	49.77 ± 3.26	6.5
Tomato	Sample 1	ND	-	ND	-
	Sample 2	ND	-	ND	-
	Sample 3	ND	-	ND	-
	Sample 4	ND	-	ND	-
	Sample 5	ND	-	ND	-

ND, Not detected. -, unavailable.

## Data Availability

Data is contained within the article or Appendix A.

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
