# Peer review of "Development of a Time-Resolved Fluorescent Microsphere Test Strip for Rapid, On-Site, and Sensitive Detection of Picoxystrobin in Vegetables"

_foods, 2024, doi:10.3390/foods13030423_

Round 1

Reviewer 1 Report

Comments and Suggestions for Authors

1. Line 79-85:

This paragraph is unclear and need to rewrite .

2. Line 231: "C18":

Write name of this material

3. Line 237: "Following TRFICA detection, GC--MS/MS MS/MS was used for verification":

The authors must mentioned to the procedure steps of GC and reference

4. Line 364: "Table 1. Addition and recovery experiment of TRFICA (n=3)Addition and recovery experiment of TRFICA (n=3).":

Do you use a statistics analysis? and the authors must add a section for statistics analysis.

5. Discussion is a poorly

6. Conclusion:

Rewrite without results,, the conclusions are not suitable.

Reviewer 2 Report

Comments and Suggestions for Authors

The comments/suggestions to the authors are appended in the word file attached herewith 

Round 2

Reviewer 1 Report

Comments and Suggestions for Authors

1. Conclusion:

the conclusions are not suitable and must rewrite without results,

Author Response

Comments 1 the conclusions are not suitable and must rewrite without results.

Response 1: Thanks for your suggestion. We will summarize and discuss the conclusions and content of the manuscript again, as follows: We have made modifications in the manuscript, as you can see on the page 14, line 452-462 of the manuscript.

Initially, we systematically investigated the hapten design, the antigen synthesis, and screened for a sensitive, specific PIC-mAb. To obtain the best detection performance, the experimental parameters of the test strips were optimized including the activation pH, antibody diluents, and amounts of antibody and TRFM probe. To address potential interferences from different vegetable matrices, we developed detection standard curves for various matrix samples. The corresponding qLOD for cucumber, green pepper, and tomato were 0.61, 0.26, and 3.44 ng/mL, respectively. Blind samples were analyzed by both the TRFICA and UP-LCMS/MS, and a good correlation between the two methods was obtained. This method could be used for rapid screening and detection of vegetable blind samples in the market, providing a means and tool for on-site, rapid, and sensitive monitoring of PIC residues in blind samples.